

# Global tropical cyclone size and intensity reconstruction dataset for 1959–2022 based on IBTrACS and ERA5 data

Zhiqi Xu[1], Jianping Guo[2*], Guwei Zhang[1], Yuchen Ye[3], Haikun Zhao[3], Haishan Chen[3]

[1]Institute of Urban Meteorology, China Metrological Administration, Beijing 100089, China

[2]State Key Laboratory of Severe Weather, Chinese Academy of Meteorological Sciences, Beijing 100081, China

[3]Key Laboratory of Meteorological Disaster, Ministry of Education (KLME)/Joint International Research Laboratory of Climate and Environment Change (ILCEC)/Collaborative Innovation Center on Forecast and Evaluation of Meteorological Disasters (CIC-FEMD), Nanjing University of Information Science and Technology, Nanjing, 210044, China

**Correspondence:** J. Guo (Email: jpguocams@gmail.com)

**Abstract.** Tropical cyclones (TCs) are powerful weather systems that can cause extreme disasters. The International Best Track Archive for Climate Stewardship (IBTrACS) dataset has been used extensively to estimate TC climatology. However, it has low data coverage, lacking intensity and outer size data for more than half of all recorded storms, and is therefore insufficient as a reference for researchers and decision makers. To fill this data gap, we reconstructed a long-term TC dataset by integrating IBTrACS and European Centre for Medium-Range Weather Forecasts Reanalysis 5 (ERA5) data. This new dataset covers the period 1959–2022, with 3 h temporal resolution. Compared to the IBTrACS dataset, it contains approximately 3–4 times more data points per characteristic. We established machine learning models to estimate the maximum sustained wind speed ($V_{max}$) and radius to maximum wind speed ($R_{max}$) in six basins for which TCs were generated using ERA5-derived 10 m azimuthal median azimuthal wind profiles as input, with $V_{max}$ and $R_{max}$ data from the IBTrACS dataset used as training data. An empirical wind–pressure relationship and six wind profile models were employed to estimate the minimum central pressure ($P_{min}$) and outer size of the TCs, respectively. Overall, this high-resolution TC reconstruction dataset demonstrated global consistency with observations, exhibiting mean biases of <1% for $V_{max}$ and 3% for $R_{max}$ and $P_{min}$ in almost all basins. The new dataset is publicly available from https://doi.org/10.5281/zenodo.12740372 (Xu et al., 2024) and significantly advances our understanding of TC climatology, thereby facilitating risk assessments and defenses against TC-related disasters.



## 1. Introduction

Tropical cyclones (TCs) are formidable weather systems accompanied by gale winds, torrential rainstorms, significant waves, and devastating storm surges, which cause extensive damage in affected regions (Gray, 1968). During the past two decades, TCs have resulted in an average annual economic loss of 29 billion dollars, affecting more than 22 million individuals (Guha-Sapir, 2017; Geiger et al., 2018). Given the considerable scale and frequency of TC-related disasters, a comprehensive understanding of TC climatology is essential for effective risk assessment, emergency planning, and community resilience enhancement.

TCs are typically characterized according to their intensity, size, location, and translation speed (Weber et al., 2014). Many studies have reported increasing TC intensity at both the basin and global scales under global warming (e.g., Webster et al., 2006; Gualdi et al., 2008; Wu et al., 2022). Vincent et al. (2014) detected a 30% increase in high-intensity TCs at the global scale. Mei and Xie (2016) demonstrated a significant correlation between TC intensification and increasing sea surface temperatures (SSTs) in East and Southeast Asia. In addition, significant increasing trends in TC intensity have been observed in the Atlantic basin over the past few decades (Walsh et al., 2016). However, assessments of the response of TC intensity to climate change are subject to uncertainty, partly due to the challenging and costly process of collecting observation data (Gualdi et al., 2008; Knutson et al., 2019). Furthermore, the movement of TCs may be significantly influenced by their size (Liu and Chan, 1999), further contributing to their destructive potential (Xu et al., 2020). Similarly, a significant increase in TC size was reported to be proportional to surface latent heat flux under warmer air and ocean temperatures (Hill and Lackmann, 2009; Radu et al., 2014). Xu et al. (2020) demonstrated that TC size increases with ocean warming, based on idealized experiments. Sun et al. (2013, 2014) discovered that TC size increases significantly as SST increases through a modeling analysis. However, the conclusions of these case studies are necessarily limited, and the relationships between TC size and climatology factors remain unclear due to the lack of historical records (Xu et al., 2020).

The International Best Track Archive for Climate Stewardship (IBTrACS) dataset is one of the most commonly used sources for TC data; it contains location, intensity, and size data for all known tropical and subtropical cyclones at a resolution



of 3 h (Knapp et al., 2010). In this dataset, maximum sustained wind speed ($V_{max}$) and minimum central pressure ($P_{min}$) are
used to quantify TC intensity (Simpson, 1974; Chavas et al., 2017; Casas et al., 2023). Among the several metrics that have
been defined to measure TC size, one of the most widely recognized is the radius to maximum wind speed ($R_{max}$, Chavas et
al., 2015; Ren et al., 2022). Radial distances from the cyclone center to locations where sustained wind speeds of 34, 50 and
64 knots (~17, 26, and 33 m/s) are observed on surface, i.e., $R_{34}$, $R_{50}$, and $R_{64}$, are also used to estimate TC size (Pérez-
Alarcón et al., 2023). However, reliable TC size and intensity estimates are available only from 1988 onwards (Demuth et al.,
2006), and post-storm analyses of wind radii, including $R_{34}$, $R_{50}$, and $R_{64}$, did not commence until 2004 (Gori et al., 2023).
Furthermore, more than half of all recorded storms lack intensity and size data, often with only location data provided even
during periods when post-storm analyses were conducted. Thus, constructing a TC climatology is an arduous task due to low
data coverage.
Machine learning has been widely used to reconstruct TC datasets. Yang et al. (2022) divided hurricane wind fields into
symmetric and asymmetric components, and proposed a downscaling model based on the XGBoost software library to
reconstruct TC structure; however, $V_{max}$ and $R_{max}$ were the model input variables. Zhuo and Tan (2023) applied deep
learning algorithms to estimate reliable TC sizes over the western North Pacific during 1981–2017, based on a homogeneous
satellite database. Li et al. (2024) proposed a transfer learning-based generative adversarial network framework to derive TC
wind fields from synthetic aperture radar images. Eusebi et al. (2024) demonstrated that a physics-informed neural network
can produce accurate reconstructions of TC wind and pressure fields by assimilating observations in a computationally efficient
manner. Nevertheless, the datasets used in these studies were generally limited to several cases or specific regions of interest,
and some are not publicly available.
By contrast, reanalysis datasets such as the fifth-generation European Centre for Medium-Range Weather Forecasts
(ECMWF) Reanalysis 5 (ERA5) dataset (Hersbach et al., 2020), the 55-year Japanese Reanalysis (Kobayashi et al., 2015), and
US National Centers for Environmental Prediction and National Centre for Atmospheric Research Reanalysis products (Kistler
et al., 2001), which combine past observations and model results through data assimilation, have unique advantages in terms



71 of data availability and spatiotemporal coverage. Previous studies have suggested that ERA5 products are among the most

72 promising reanalysis data sources in terms of representing TC outer size and structure, due to their relatively fine horizontal

73 grid spacing (Bian et al., 2021; Pérez-Alarcón et al., 2023; Dulac et al., 2024). The reconstruction of TC proxies using ERA5

74 data has been demonstrated to be a viable approach (Yeasmin et al., 2023). Nevertheless, due to horizontal resolution limits

75 and conservative physics parameterizations, reanalysis products have exhibited large underestimation and overestimation of

76 TC $V_{max}$ and $R_{max}$ values, respectively (Hatsushika et al., 2006; Schenkel and Hart, 2012). Thus, despite the substantial

77 body of research reconstructing the outer sizes and proxies of TCs using ERA5 data (Bian et al., 2021; Gori et al., 2023; Pérez-

78 Alarcón et al., 2023), studies based on its relatively accurate TC intensity data are lacking.

79 In this study, we exploited the advantages of the IBTrACS and ERA5 datasets to generate a new TC dataset containing

80 all characteristics of TCs. Given the high degree of accuracy demonstrated by the ERA5 data in capturing TC structures, we

81 employed ERA5-derived azimuthal median azimuthal wind profiles in conjunction with a machine learning model to reduce

82 the bias observed in the $V_{max}$ and $R_{max}$ of TCs between the ERA5 and IBTrACS datasets. In addition, we modeled six TC

83 radial wind profiles to compute $R_{34}$, $R_{50}$, and $R_{64}$. The resulting long-term TC reconstruction dataset covering the period

84 1959–2022 is anticipated to facilitate future TC climatology research. The generated dataset is approximately 3–4 times larger

85 than the IBTrACS dataset in terms of the number of records per characteristic.

86 In the subsequent sections, we describe the IBTrACS and ERA5 datasets and the methodology used to create the novel

87 TC reconstruction dataset. The findings are reported and discussed in comparison with IBTrACS data according to a

88 comprehensive set of statistical metrics. Finally, we consider the potential applications of the reconstructed TC dataset.

89 **2. Data**

90 **2.1 IBTrACS data**

91 Data on TC tracks, intensity, and size were obtained from the IBTrACS (version 4r01, which is a unified dataset containing

92 track estimates for all TC basins with a 3 h temporal resolution, based on data produced by tropical warning centers. As the

93 TC $R_{max}$ data from all main TC basins were accessible from U.S. agencies, we employed these data and excluded the irregular





time steps. All TC events in all basins were used, except for those over the South Atlantic, where TC generation is insufficient.
A comprehensive overview of the recorded TC characteristics is presented in Table 1. The IBTrACS dataset encompasses a
total of 7,552 TCs on a global scale, spanning the period 1959–2022, corresponding to 423,296 individual time points. However,
only 125,477 $V_{max}$, 142,430 $P_{min}$, and 94,415 $R_{max}$ values were recorded. TC tracks and $V_{max}$ data extracted from the
IBTrACS dataset are presented in Fig. 1.

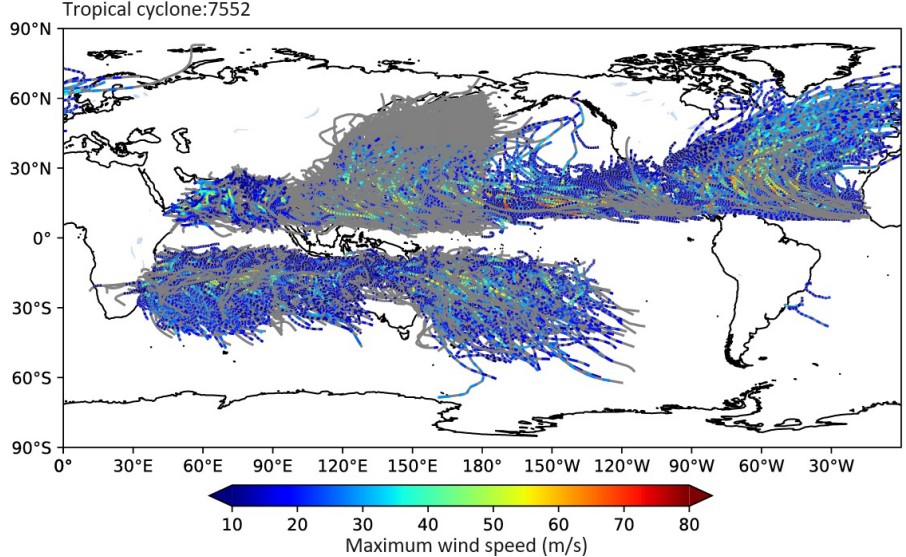

**Figure 1: Overview of the tracks and 10-m maximum wind speeds of tropical cyclones in IBTrACS dataset. Grey lines represent the**
**unrecorded wind speeds.**
**Table 1: Basic information on the number of recorded tropical cyclone characteristics from 1959 to 2022 recorded in IBTrACS.**

| Basin | Time point | $V_{max}$ | $P_{min}$ | $R_{max}$ | $R_{34}$ | $R_{50}$ | $R_{64}$ |
|---|---|---|---|---|---|---|---|
| Western Pacific | 152362 | 26604 | 61018 | 28715 | 19340 | 10641 | 7149 |
| North Atlantic | 55679 | 28310 | 21409 | 18161 | 14961 | 7630 | 4212 |
| North Indian | 24101 | 5481 | 5476 | 4281 | 2354 | 1029 | 614 |
| South Indian | 86790 | 23935 | 24468 | 16367 | 10697 | 5108 | 2977 |
| South Pacific | 45189 | 12322 | 12467 | 7169 | 4827 | 2577 | 1521 |
| Eastern Pacific | 59175 | 28825 | 17592 | 19722 | 12283 | 6482 | 3986 |
| Global | 423296 | 125477 | 142430 | 94415 | 64462 | 33467 | 20459 |




**2.2 ERA5 data**

ERA5 is the latest ECMWF reanalysis, following a decade of developments in model physics, core dynamics, and data assimilation (Hersbach et al., 2020). We utilized the main ERA5 dataset for the period 1959–2022 to estimate the track, intensity, and size of each TC. The spatial resolution of the ERA5 dataset is 0.25° × 0.25°, with a temporal resolution of 3 h, aligning with that of the IBTrACS dataset. Pre-1959 ERA5 back-extension data were not adopted, as some TCs in these data exhibited unrealistically high levels of tension (Bell, 2021). Notably, despite the higher uncertainty associated with TC intensity data derived from ERA5 for the pre-satellite time period (1959–1978), comparisons of TC intensity pre- and post-1979 revealed similar climatological distributions for both TC groups in all basins (Fig. S1). We employed 10 m surface meridional and latitudinal wind speeds to obtain 10 m azimuthal–mean azimuthal wind profiles for TCs. The sea level pressure (SLP) was utilized to provide environmental pressure data for computing the TC central pressure. Parameters including the SLP; relative vorticity at 700, 850, and 925 hPa; and geopotential height at 700 and 850 hPa were derived from the ERA5 data to identify TC centers.

**3. Methodology**

**3.1 TC center identification and azimuthal wind profile estimation**

TC centers in the ERA5 data were identified based on the method of Schenkel (2017). The position of each TC within the reanalysis grid was initially ascertained utilizing the IBTrACS position as a first guess. To remove uncertainties associated with TC centers in the reanalysis data, the centroid of six reanalysis variables (SLP; relative vorticity at 700, 850, and 925 hPa; and geopotential height at 700 and 850 hPa) was averaged over the grid near the first guess position to adjust the position of the estimated reanalysis TC center.

Azimuthal wind profiles based on the ERA5 data were estimated as described by Chavas and Vigh (2014). First, estimated environment wind fields, which were calculated as 0.55 of the TC translation vectors rotated 20° counterclockwise (Lin and Chavas, 2012) were subtracted from the meridional and latitudinal wind speeds. TC translation vectors were determined according to the TC positions at the next and current time points in the IBTrACS data. Next, the 10 m surface meridional and



latitudinal wind fields were interpolated to a TC-centered polar coordinate. In contrast to the method of Chavas and Vigh, we
did not exclude grid points over land to obtain the TC intensity after landfall. Then, the parameter $\mathcal{X}$, defined as the normalized
average magnitude of all vectors from the TC center to each grid point included at a specified radius (Chavas and Vigh, 2014)
was employed to remove asymmetrical radial bins by excluding radial bins with $\mathcal{X} > 0.5$. Finally, the TC 10 m azimuthal–
mean azimuthal wind profiles were calculated as changes in wind speed with distance from the TC center, with grid points
spaced at 10 km intervals. The ERA5-derived TC $V_{max}$ ($V_{max\_ERA5}$) and $R_{max}$ ($R_{max\_ERA5}$) were obtained from the wind
profiles.
**3.2 Machine learning model for reconstructing TC $V_{max}$ and $R_{max}$ from ERA5 data**
As shown in Fig. 2, there were discernible biases in all six TC basins between the ERA5- and IBTrACS-derived $V_{max}$ and
$R_{max}$ values. The biases of $V_{max}$ were less dependent on the basin, suggesting the systematic underestimation of $V_{max}$ by
the ERA5 data. In contrast, biases were more pronounced for larger $V_{max}$ values, with underestimation detected for wind
speeds exceeding 20 and 30 m/s for Saffir–Simpson categories 1–2 and 3–5, respectively, in all six basins. Notably, this bias
even exceeded 40 m/s for Saffir–Simpson categories 3–5 in the East Pacific basin. In addition, ERA5-derived results
overestimated $R_{max}$ by >15 km in all basins, and by >80 km in the West Pacific basin. The large biases produced by ERA5
motivated us to establish a new TC dataset that is more consistent with observations.

Previous studies have indicated that despite the modesty of ERA5-derived TC intensity data, the ERA5 dataset accurately

depicts TC structural alterations (Bian et al., 2021). Therefore, we used the TC 10 m azimuthal–mean azimuthal wind speed
at radial distances from 0 to 1000 km, at 10 km intervals, as a parameter to estimate $V_{max}$ in each basin. The parameters also
included the TC translation speed, given that the IBTrACS $V_{max}$ data ($V_{max\_IB}$) represent a combination of the environmental
and TC wind fields. After testing several machine learning models, including an artificial neural network, convolutional neural
network, support vector regressor, multilayer perceptron regression, and random forest (RF) algorithms, we found that RF
provided the most robust predictions. Therefore, an RF regressor was developed to predict reconstructed $V_{max}$ ($V_{max\_RC}$), as
follows:



$$V_{max\_RC} = RF(V_0, V_{10}, V_{20}, \ldots, V_{1000}, V_{TS}) \tag{1}$$

where $RF$ and $V_{TS}$ are the RF regressor and TC translation speed, respectively, and $V_0, V_{10}, V_{20}, \ldots, V_{1000}$ refer to the 10 m

azimuthal–mean azimuthal wind speeds at radial distances from 0 to 1000 km.

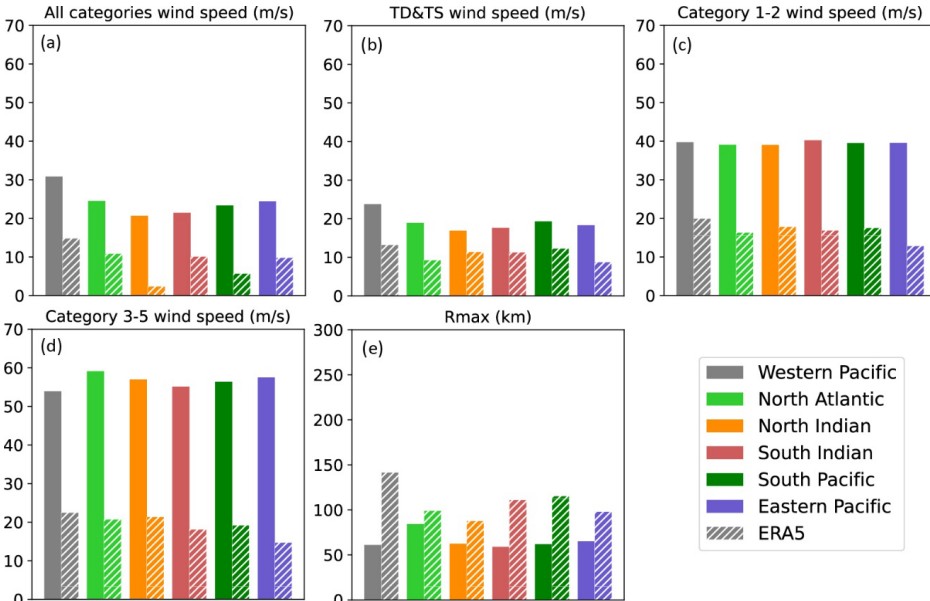

**Figure 2: Bar charts for comparing the mean value of the 10-m maximum wind speeds and the radii to maximum winds. Each of the colors indicates a different basin. Solid and dashed bars represent IBTrACS and ERA5-derived data.**

Similarly, variation in radial distance with azimuthal wind speed was used to estimate $R_{max}$ in the six basins. After

testing several machine learning models, the RF regressor was utilized to predict the reconstructed $R_{max}$ ($R_{max\_RC}$), as

follows:

$$R_{max\_RC} = RF(R_0, R_{0.01}, R_{0.02}, \ldots, R_1) \tag{2}$$

where $R_0, R_{0.01}, R_{0.02}, \ldots, R_1$ represent the radial distances at which normalized wind speeds range from 0 to 1, at an interval

of 0.01. In the RF models, hyperparameters including the maximum tree depth, minimum leaf samples, minimum sample splits,

and maximum leaf nodes were determined by randomized searches. The dataset, made up of the input array and learning target,

was randomly divided into two subsets, with 75% allocated for training and the remaining 25% for validation, following the

methods of previous studies (e.g., Breiman, 2001; Guo et al., 2024). Training data for the entire period (1959–2022) were



incorporated into the model training process. Model performance was evaluated using a comprehensive set of statistical metrics,
including mean error, mean absolute error, root mean square error (RMSE), and correlation coefficients.
**3.3 Empirical wind speed–pressure relationship for determining $P_{min}$**
The conversion between $V_{max}$ and $P_{min}$ at a given time point during a TC was modeled using the empirical wind–pressure
relationship (Atkinson and Holliday, 1977; Harper, 2002), as follows:
$$V_{max} = a(P_{env} - P_{min})^b \tag{3}$$
where $P_{env}$ is the environmental pressure obtained from the mean SLP for the TC center location 1–10 days earlier based on
the ERA5 data, following the method of Bloemendaal et al. (2020); $a$ and $b$ were estimated in each basin using a nonlinear
least squares approach, based on $V_{max}$ and the corresponding $P_{min}$ of the IBTrACS dataset. $V_{\max\_RC}$ was input into the
fitted Eq. (3) to obtain the reconstructed $P_{min}$ ($P_{min\_RC}$).
**3.4 TC radial wind profile models for computing $R_{34}$, $R_{50}$, and $R_{64}$**
Previous studies have developed TC radial wind profile models for estimating TC structures (e.g., Pérez-Alarcón et al., 2021).
After obtaining the reconstructed $V_{max}$ and $R_{max}$, six widely used wind field models (Holland, 1980; DeMaria, 1987;
Willoughby et al., 2006; Emanuel and Rotunno, 2011; Frisius and Scgönemann, 2013; Chavas et al., 2015), were used to
estimate the reconstructed TC $R_{34}$, $R_{50}$, and $R_{64}$ ($R_{34\_RC}$, $R_{50\_RC}$, and $R_{64\_RC}$).
The wind profile model proposed by Holland (1980) was formulated as follows:
$$V(r) = V_{max} \sqrt{\left(\frac{R_{max}}{r}\right)^b e^{1-\left(\frac{r}{R_{max}}\right)^{-b}}} \tag{4}$$
where $V$ is the wind speed at distance $r$ from the TC center, and $b$ = 2, according to Kowaleski and Evans (2016).
The model developed by DeMaria (1987) was formulated as follows:
$$V(r) = V_{max} \left(\frac{R_{max}}{r}\right) e^{\frac{\frac{1}{c} - \frac{1}{c}\left(\frac{r}{R_{max}}\right)^c}{d}} \tag{5}$$
where $c$ = 0.63 and $d$ = 1, following Kowaleski and Evans (2016).
The model proposed by Willoughby et al. (2006; hereinafter, W06) was formulated as follows:



$$V(r) = \begin{cases} V_{max}\left(\frac{r}{R_{max}}\right)^n, & 0 \leq r \leq R_1 \\ V_i(1-w) + V_0 w, & R_1 \leq r \leq R_2 \\ V_{max} e^{-\frac{r-R_{max}}{X_1}}, & R_2 \leq r \end{cases}$$ (6)
where $V_i$ and $V_0$ are the tangential wind components in the eye and beyond the transition zone, respectively, and $w$, $X_1$, and
$n$ are the weight function, exponential decay length in the outer vortex, and power law exponent within the eye, respectively.
The model proposed by Emanuel and Rotunno (2011) was formulated as follows:
$$V(r) = \frac{2r(R_{max}V_{max} + 0.5fR_{max}^2)}{R_{max}^2 + r^2} - \frac{fr}{2}$$ (7)
where $f$ is the Coriolis parameter.
The model developed by Frisius and Scgönemann (2013) was formulated as follows:
$$V(r) = V_{max}\frac{r}{R_{max}}\left[\frac{2(\frac{R_{max}}{r})^2}{2-(\frac{C_H}{C_d})[1-(\frac{r}{R_{max}})^2]}\right]^{\frac{1}{2}-\frac{C_H}{C_d}} - \frac{fr}{2}$$ (8)
where $C_H$ and $C_D$ are the surface enthalpy transfer and drag coefficients, respectively, and $\frac{C_H}{C_d} = 1$, according to Frisius and
Scgönemann (2013).
The model proposed by Chavas et al. (2015; hereinafter, CLE15) was formulated as follows:
$$\left(\frac{M_{inner}}{M_m}\right)^{2-\frac{C_k}{C_d}} = \frac{2(\frac{r}{R_{max}})^2}{2-(\frac{C_k}{C_d})+(\frac{C_k}{C_d})(\frac{r}{R_{max}})^2}$$ (9)
$$\frac{\partial M_{outer}}{\partial r} = \frac{C_d(rV)^2}{0.001(r_o^2 - r^2)}$$
where $M_{inner}$, $M_{outer}$, and $M_m$ are the angular moment of the inner and outer wind regimes and at $R_{max}$, respectively; and
$C_k$ and $C_d$ are the exchange surface enthalpy and momentum coefficients, respectively.
The performance of each profile model was evaluated by comparing $R_{34}$, $R_{50}$, and $R_{64}$ estimates with those recorded
in the IBTrACS dataset. The optimal model was selected to generate reconstructed $R_{34}$, $R_{50}$, and $R_{64}$, as described in detail
in Section 4.
**3.5 Flowchart for optimal wind profile model selection**
After identifying the TC center, we used an RF approach to estimate $V_{max}$ and $R_{max}$ based on the ERA5-derived TC 10 m
azimuthal–mean azimuthal wind profiles. Next, the parameters of the empirical wind–pressure relationship were estimated,



and TC $P_{min}$ values were computed. Finally, the TC $R_{34}$, $R_{50}$, and $R_{64}$ were derived by selecting the optimal wind profile
model from among the six widely used models. The overall methodology is illustrated in Fig. 3.

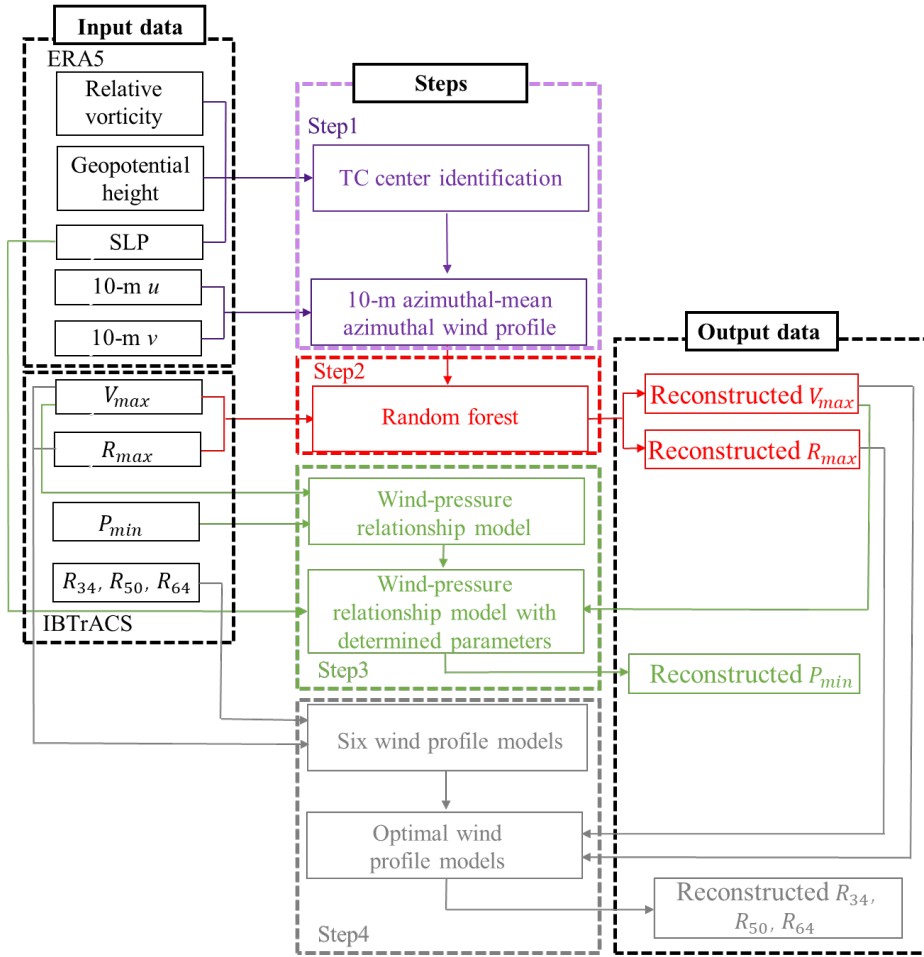


**Figure 3: Flowchart with the tropical cyclone center identification and wind profiles extracted from ERA5 (Step 1; in purple), the**
**10-m maximum wind speeds and radii to maximum winds estimated by random forest model (Step 2; in red), the minimum central**
**pressure estimated by empirical wind-pressure relationship (Step 3; in green), and the out size estimated by wind profile models**
**(Step 4; in grey).**
**4. Results and Discussion**
The accuracy of the $V_{max\_RC}$ model results was evaluated according to various statistical metrics based on the testing datasets
(Fig. 4), as prescribed by Breiman (2001). The $V_{max\_RC}$ data were strongly correlated with observations, with correlation
coefficients exceeding 0.98 for all six basins. The RMSE values for the West Pacific, North Atlantic, North and South Indian
Ocean, and South and East Pacific basins were 2.60, 4.09, 1.33, 3.25, 3.73, and 5.05 m/s, respectively. Compared to $V_{max\_ERA5}$,





the reconstruction provided a reduction in the mean absolute bias of over 10 m/s in most basins, with a further reduction of
19.62 m/s in the East Pacific basin, as described in detail in Table 2. The model was more effective at reducing biases between
ERA5-derived results and observations for larger $V_{max}$ values. Furthermore, given the high influence of ENSO on TC
intensity (Chu, 2024), the accuracy of $V_{max\_RC}$ was evaluated for moderate to strong El Niño and La Niña years (Fig. S2 and
S3). A high degree of correlation coefficients (>0.97) and low RMSE values (<5m/s) were observed between $V_{max\_RC}$ and
$V_{max}$ in all six basins during ENSO years. These metrics clearly demonstrate the superior accuracy of $V_{max\_RC}$ and its
reduced bias compared to $V_{max\_ERA5}$.

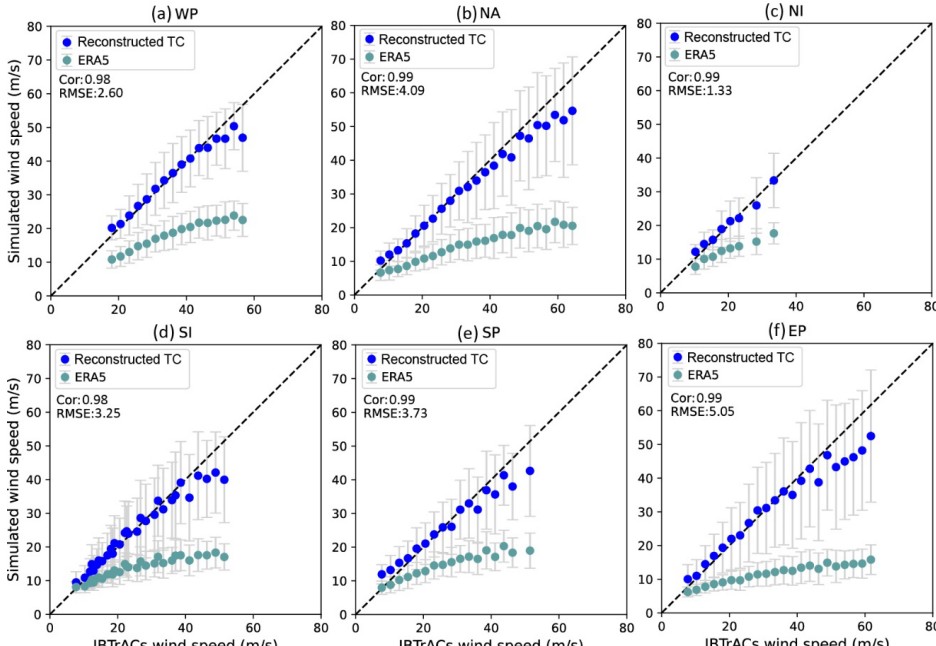

**Figure 4: Comparison between value-averaged maximum wind speeds ($V_{max}$) from ERA5-derived and reconstructed (ERA5 +**
**Random forest) data and IBTrACS maximum wind speeds for tropical cyclones in (a) Western Pacific, (b) North Atlantic, (c) North**
**Indian, (d) South Indian, (e) South Pacific and (f) Eastern Pacific basins. Grey lines represent the error bar, given as one standard**
**deviation from the mean. The values with sample sizes less than 30 in IBTrACS were excluded.**





Table 2: Basic information on evaluation indices for $V_{max}$. ME, mean errors; MAE, mean of the absolute bias; RMSE, root mean
square error; CE, correlation coefficients.

|  | ME (m/s) | MAE (m/s) | RMSE (m/s) | CE |
|---|---|---|---|---|
| Global_ERA5 | 16.73 | 16.80 | 21.70 | 0.92 |
| Global_Reconstructed | 2.82 | 2.83 | 4.34 | 0.99 |
| WP_ERA5 | 18.93 | 18.93 | 20.54 | 0.97 |
| WP_Reconstructed | 0.56 | 1.63 | 2.60 | 0.98 |
| NA_ERA5 | 21.03 | 21.03 | 24.46 | 0.98 |
| NA_Reconstructed | 2.38 | 2.82 | 4.09 | 0.99 |
| NI_ERA5 | 7.74 | 7.74 | 8.96 | 0.98 |
| NI_Reconstructed | -0.25 | 1.11 | 1.33 | 0.99 |
| SI_ERA5 | 12.39 | 12.41 | 15.61 | 0.93 |
| SI_Reconstructed | 0.71 | 2.17 | 3.25 | 0.98 |
| SP_ERA5 | 13.71 | 13.73 | 16.67 | 0.96 |
| SP_Reconstructed | 1.19 | 2.70 | 3.73 | 0.99 |
| EP_ERA5 | 23.09 | 23.09 | 26.86 | 0.97 |
| EP_Reconstructed | 2.36 | 3.47 | 5.05 | 0.99 |

We similarly evaluated the accuracy of $R_{max\_RC}$ for the six basins based on the testing datasets (Fig. 5). Correlation
coefficients between $R_{max\_RC}$ and $R_{max}$ recorded in IBTrACS ($R_{max\_IB}$) exceeded 0.9, indicating strong correlation
between the reconstructed results and observations. Moreover, the RMSEs for the West Pacific, North Atlantic, North and
South Indian Ocean, and South and East Pacific basins were 20.80, 31.47 10.48, 16.51, 15.11, and 24.75 km, respectively.
Importantly, $R_{max\_ERA5}$ exhibited a large deviation from observations, exceeding 300 km at very low $R_{max\_IB}$ values.
Therefore, for clarity, the $R_{max\_ERA5}$ data are not shown with the reconstructed TC results in Fig. 5. The mean absolute bias
exhibited a reduction of 39.57 km on a global scale, with a further reduction of over 59.37 km in the South Indian Ocean basin,
as described in detail in Table 3. Although the $R_{max\_RC}$ data slightly overestimated observations at low $R_{max\_IB}$ values and
underestimated observations at high $R_{max\_IB}$ values, they greatly reduced biases compared to the $R_{max\_ERA5}$ data, and thus
produced superior predictions for all six basins.



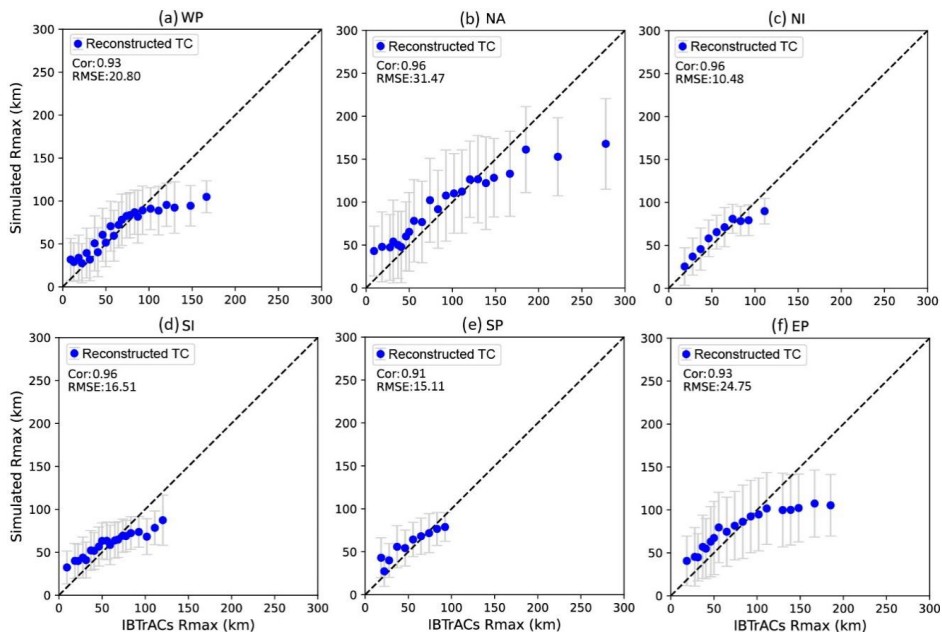

**Figure 5. Similar to Figure 4, but for radii to maximum winds ($R_{max}$).**
**Table 3: Similar to Table 2, but for $R_{max}$.**

|  | ME (km) | MAE (km) | RMSE (km) | CE |
|---|---|---|---|---|
| Global$_{ERA5}$ | -41.64 | 15.92 | 67.66 | 0.44 |
| Global$_{Reconstructed}$ | 1.37 | 55.49 | 22.19 | 0.94 |
| WP$_{ERA5}$ | -56.43 | 58.31 | 69.86 | 0.75 |
| WP$_{Reconstructed}$ | 1.32 | 14.93 | 20.80 | 0.93 |
| NA$_{ERA5}$ | -7.79 | 54.25 | 64.59 | 0.37 |
| NA$_{Reconstructed}$ | 4.05 | 21.44 | 31.47 | 0.96 |
| NI$_{ERA5}$ | -28.95 | 29.39 | 33.75 | 0.96 |
| NI$_{Reconstructed}$ | -2.30 | 9.65 | 10.48 | 0.96 |
| SI$_{ERA5}$ | -73.40 | 73.48 | 88.39 | 0.74 |
| SI$_{Reconstructed}$ | -1.50 | 14.11 | 16.51 | 0.96 |
| SP$_{ERA5}$ | -52.42 | 52.99 | 61.95 | 0.90 |
| SP$_{Reconstructed}$ | -3.21 | 12.09 | 15.11 | 0.91 |
| EP$_{ERA5}$ | -24.31 | 47.83 | 56.59 | -0.02 |
| EP$_{Reconstructed}$ | 6.91 | 18.83 | 24.75 | 0.93 |





$P_{min\_RC}$ was computed based on an empirical wind–pressure relationship. $V_{max\_IB}$ and the corresponding $P_{min}$
recorded in IBTrACS ($P_{min\_IB}$) were also employed in the reconstruction, and $P_{env}$ was obtained from the ERA5 dataset,
following the method of Bloemendaal et al. (2020). Related parameters were estimated through nonlinear fitting; the results
are shown in Fig. 6. For the West Pacific, North Atlantic, North and South Indian Ocean, and South and East Pacific basins,
we used *a* values of 0.118, 0.051, 0.259, 0.184, 0.325, and 0.073 and *b* values of 1.67, 1.692, 1.402, 1.507, 1.371, and 1.651,
respectively, in Eq. (3).

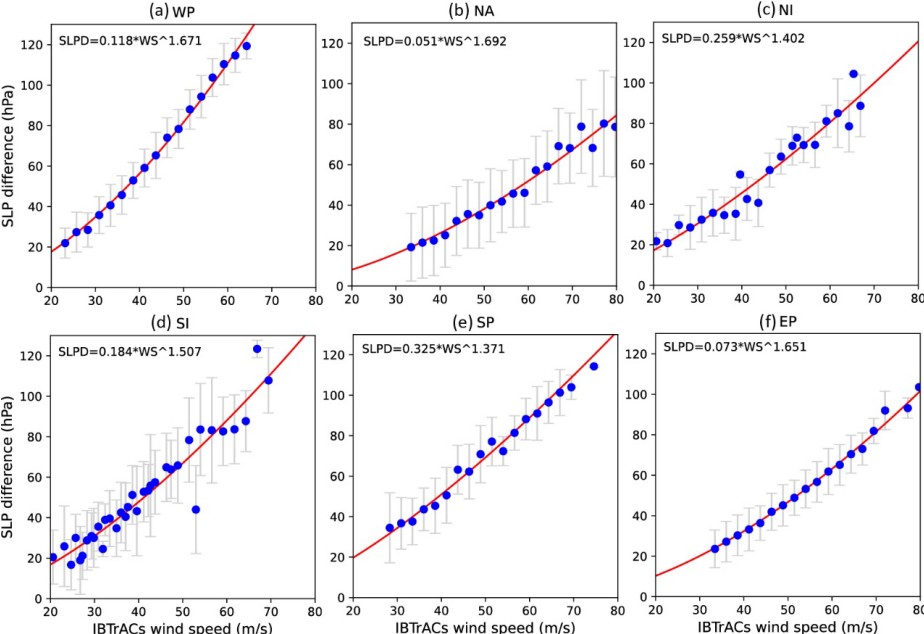

**Figure 6: Similar to Figure 4, but for non-linear regression analyses between value-averaged IBTrACS maximum wind**
**speeds and sea level pressure difference (SLPD).**
The mean and standard deviation values of various TC characteristics based on the testing datasets are plotted in Fig. 7
to compare the overall performance of the model in reconstructing TCs. Mean biases in $R_{max}$ and $P_{min}$ between the
reconstructed TC and IBTrACS datasets were both <3% in most basins, providing compelling evidence that the predictions
were in good agreement with observations. In contrast to those over the sea, the reconstructed landfall TC $V_{max}$ and $R_{max}$
data were overestimated and underestimated in most basins, respectively, likely due to the decay of TC wind speeds after



landfall, which is not considered in the RF-based models. Despite these differences, biases remained within 5% in most basins,
indicating that the reconstructed landfall TC characteristics were closely aligned with those in the IBTrACS dataset.

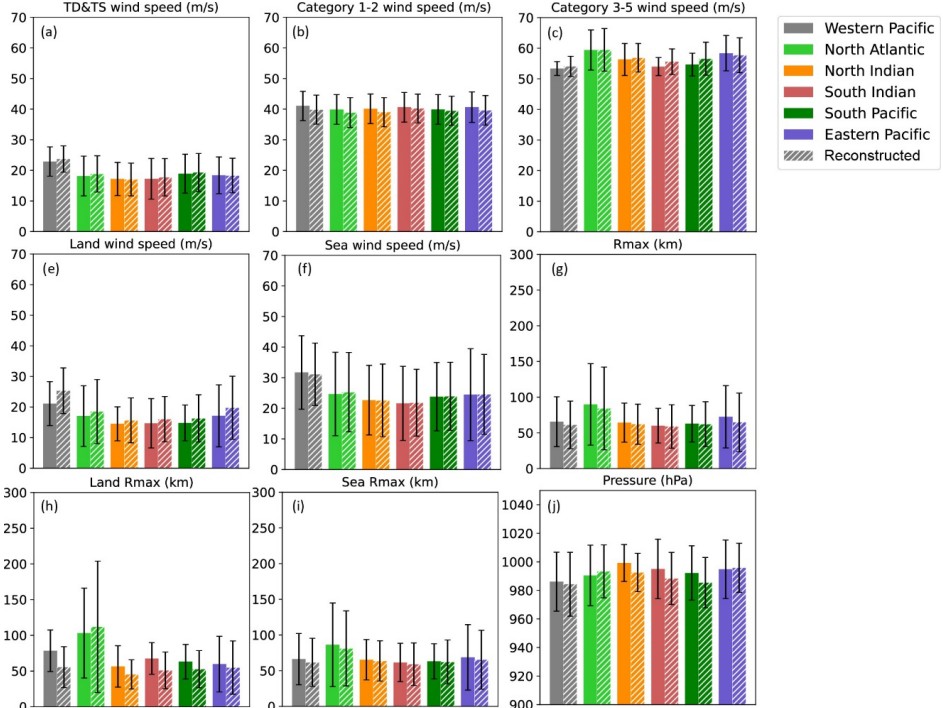


**Figure 7: Bar charts for comparing the mean value of the different tropical cyclone characteristics. Each of the colors indicates a**
**different basin. Solid and dashed bars represent IBTrACS and reconstructed tropical cyclone data.**

After obtaining the reconstructed TC intensity dataset, six widely used models were used to estimate $R_{34\_RC}$, $R_{50\_RC}$, and
$R_{64\_RC}$. We conducted a comparative analysis of the model-derived results and observations to determine which radial wind
profile estimate more closely approximated the TC outer radius, based on various statistical metrics (Table S1–S6). The W06
model results exhibited strong correlation, low RMSE, and low absolute mean error for all basins except the North Atlantic,
whereas the CLE15 model performed better for $R_{34\_RC}$ in the North Atlantic basin. Therefore, we used W06 to forecast
$R_{34\_RC}$, $R_{50\_RC}$, and $R_{64\_RC}$ for the West Pacific, North and South Indian Ocean, and South and East Pacific basins, whereas
for the North Atlantic basin, we used CLE15 to predict $R_{34\_RC}$ and W06 to predict $R_{50\_RC}$ and $R_{64\_RC}$. The correlation
coefficients were >0.75 for three outer size metrics in most basins (Table 4).



**Table 4: Similar to Table 2, but for $R_{34}$, $R_{50}$ and $R_{64}$.**

|          | Optimal profile | ME (km) | MAE (km) | RMSE (km) | CE |
|----------|-----------------|---------|----------|-----------|----|
| WP $_{R34}$ | W06 | -24.79 | 46.75 | 64.54 | 0.89 |
| WP $_{R50}$ | W06 | -14.60 | 26.00 | 33.27 | 0.82 |
| WP $_{R64}$ | W06 | -14.14 | 18.28 | 22.71 | 0.78 |
| NA $_{R34}$ | CLE15 | -25.19 | 53.00 | 78.77 | 0.87 |
| NA $_{R50}$ | W06 | -11.58 | 32.71 | 57.39 | 0.84 |
| NA $_{R64}$ | W06 | 2.67 | 18.52 | 30.37 | 0.87 |
| NI $_{R34}$ | W06 | -23.19 | 31.19 | 41.59 | 0.74 |
| NI $_{R50}$ | W06 | -14.66 | 20.49 | 25.69 | 0.63 |
| NI $_{R64}$ | W06 | -11.63 | 16.62 | 21.17 | 0.62 |
| SI $_{R34}$ | W06 | 3.57 | 45.71 | 56.68 | 0.74 |
| SI $_{R50}$ | W06 | 14.35 | 29.69 | 36.18 | 0.46 |
| SI $_{R64}$ | W06 | 9.68 | 18.54 | 21.57 | 0.43 |
| SP $_{R34}$ | W06 | -5.00 | 33.51 | 46.25 | 0.83 |
| SP $_{R50}$ | W06 | 11.75 | 21.53 | 27.25 | 0.77 |
| SP $_{R64}$ | W06 | 12.75 | 15.60 | 18.56 | 0.77 |
| EP $_{R34}$ | W06 | 32.25 | 44.43 | 51.31 | 0.81 |
| EP $_{R50}$ | W06 | 27.19 | 31.77 | 36.61 | 0.68 |
| EP $_{R64}$ | W06 | 18.74 | 21.66 | 25.24 | 0.51 |

The ERA5 dataset was used to derive parameters characterizing TC intensity and size in creating the TC reconstruction
dataset. Then these parameters were subjected to a machine learning algorithm to produce more accurate data. Notably, the
TC intensity and size reconstructions developed in this study may be influenced by limitations and uncertainties inherent to
the IBTrACS and ERA5 datasets. The RF models were unable to differentiate between landfall and offshore TCs due to the
limited data available concerning landfall TCs in the IBTrACS dataset, which resulted in higher $V_{max}$ and lower $R_{max}$ values
for landfall TCs. Additionally, $R_{34}$, $R_{50}$ and $R_{64}$ were estimated using wind profile models rather than RF models due to
the paucity of relevant data, which resulted in a lower level of accuracy than for these TC characteristics. Moreover, there was
some dependency between the reconstructed and IBTrACS-derived $R_{max}$ values, likely due to the insufficient spatial
resolution of the ERA5 dataset. Besides, TC positions in the IBTrACS data exhibited some degree of inaccuracy during the
pre-satellite time period. Notwithstanding these limitations, the TC reconstruction dataset exhibited a markedly high degree of
accuracy and extensive spatiotemporal coverage. Basic information on the reconstructed TC data is presented in Table 5.



**Table 5: Basic information on the number of recorded tropical cyclone characteristics from 1959 to 2022 recorded in**
**reconstructed data.**

| Basin | $V_{max}$ | $P_{min}$ | $R_{max}$ | $R_{34}$ | $R_{50}$ | $R_{64}$ |
|---|---|---|---|---|---|---|
| Western Pacific | 152208 | 152208 | 152208 | 127668 | 39659 | 24302 |
| North Atlantic | 55608 | 55608 | 55608 | 31829 | 19106 | 11719 |
| North Indian | 24047 | 24047 | 24047 | 4614 | 1840 | 1039 |
| South Indian | 86606 | 86606 | 86606 | 35768 | 18500 | 10395 |
| South Pacific | 45112 | 45112 | 45112 | 23312 | 10547 | 5454 |
| Eastern Pacific | 59112 | 59112 | 59112 | 33772 | 19214 | 13026 |
| Global | 422693 | 422693 | 422693 | 256963 | 108866 | 65935 |

**5. Data and Code availability**
All data have been published in the form of CSV files, and are made publicly available through Zenodo repository with the
address: https://doi.org/10.5281/zenodo.12740372 (Xu et al., 2024). ERA5 data can be publicly accessible at
https://doi.org/10.24381/cds.bd0915c6 (Hersbach et al., 2023a) and https://doi.org/10.24381/cds.adbb2d47 (Hersbach et al.,
2023b). IBTrACS data is accessible at https://doi.org/10.25921/82ty-9e16 (Gahtan et al., 2024). The processing codes can be
made available upon request to the corresponding author. This study provides a detailed description of the TC size and intensity
reconstruction dataset, which includes the maximum sustained wind speed, the radius to maximum wind speed, the minimum
central pressure and the radii to locations with sustained wind speeds of 34, 50, and 64 knots during 1959–2022.
**6. Conclusion**
The considerable number of unrecorded TC characteristics in the IBTrACS dataset and large biases inherent in the ERA5
dataset prompted us to generate a long-term TC reconstruction dataset. We constructed the dataset by integrating TC
characteristics from the IBTrACS and ERA5 datasets using RF-based models, an empirical wind–pressure relationship, and
six wind profiles for the period 1959–2022. The TC reconstruction dataset is approximately 3–4 times larger than the IBTrACS
dataset in terms of data points per characteristic, with much higher data accuracy than shown for ERA5-derived results.
Six TC characteristics were examined to evaluate the reconstructed dataset. A comparison of maximum sustained wind
speeds between the IBTrACS and reconstructed TC datasets revealed that the latter underestimated observation data by
approximately 2.82 m/s, which is a considerably smaller bias than that shown by the ERA5 dataset (16.73 m/s) on a global



scale. For the radius to maximum wind speed ($R_{max}$), the mean error and RMSE decreased markedly, from −41.64 and 67.66
km (IBTrACS $R_{max}$ − ERA5 $R_{max}$) to 1.37 and 22.19 km (IBTrACS $R_{max}$ − reconstructed $R_{max}$), respectively. In
addition, the correlation coefficient for $R_{max}$ between the IBTrACS and ERA5 datasets was 0.44, which increased to 0.94
between the IBTrACS and TC reconstruction datasets. The mean bias in minimum central pressure between the IBTrACS and
reconstructed TC datasets was <3% in most basins. Six wind profile models were used to compute the radii to locations with
sustained wind speeds of 34, 50, and 64 knots ($R_{34}$, $R_{50}$, and $R_{64}$), and the selected wind profile models (CLE15 for $R_{34}$ in
the North Atlantic, W06 for others) showed good estimates for TC outer sizes, with correlation coefficients > 0.75 for three
outer size metrics in most basins. Overall, the TC reconstruction dataset agreed closely with the IBTrACS data in terms of TC
intensity and size.
In conclusion, the TC reconstruction dataset may prove invaluable for advancing our understanding of TC climatology,
thereby facilitating risk assessments and defenses against TC-related disasters. The future availability of reanalysis data with
finer spatial resolution and longer temporal coverage, such as the in-progress ERA6, will facilitate the creation of more accurate
TC reconstructions with longer time spans using the methods presented in this study.

**Author Contributions.** ZX, JG and GZ wrote the first draft of the manuscript. ZX, JG and YY developed the model code and
conducted scientific analyses. All authors contributed to the writing and the editing of the manuscript.
**Competing interests.** The contact author has declared that none of the authors has any competing interests.
**Acknowledgements.** This work was financially supported by the National Natural Science Foundation of China
(NSFC42205040 and NSFC42205170), and Youth Innovation Team of China Meteorological Administration (No.
CMA2024QN14).



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
