# Peer review of "Global tropical cyclone size and intensity reconstruction dataset for"

_Earth System Science Data, 2024_

## Author Comment (AC1)

**Responses to comments of Reviewer #1**

We wish to express our great appreciation for your review and constructive comments. In the revised manuscript, we have made substantial changes to strengthen its readability, and incorporated all your comments. In the response below, we address each of these comments individually. Your comments are *italicized* and our responses follow immediately. Changes to the manuscript text are included in blue.

*The manuscript presents a new global tropical cyclone dataset that integrates the IBTrACS and ERA5 reanalysis data to reconstruct key TC characteristics like Vmax, Rmax, and Pmin. The authors use random forest algorithm to reduce biases in the ERA5-derived characteristics, enhancing the data availability and spatiotemporal coverage of the best track dataset. This manuscript demonstrates a certain level of innovation and scientific value, and it is generally well-organized. I recommend accepting the manuscript with minor revisions in the following.*

*1. The approach of combining IBTrACS and ERA5 data using machine learning like Random Forest models appears to be well-justified based on the reported improvements in bias reduction. However, it would be helpful to provide more details about the selection process for the RF model, particularly in comparison with other models that were tested but not selected.*

**Response:** Thank you for this good comment. We have supplemented the details regarding the selection process for the RF model in the section 3.2, and have extensively rewritten most parts of this section. For the revised comparisons among different models, please see lines 162-170, Table 2, lines 181-183 and Table 3.

[lines 162-170]:
   We optimize the machine learning models by Randomized Search Cross-Validation with mean square error as the loss function using Python. The models include a random forest (RF) algorithm, artificial neural network (ANN), convolutional neural network, support vector regressor, and multivariate linear regression (Table 2). In the above-mentioned models, we incorporate data for the entire period (1959–2022) into the model training process. We randomly divide the dataset, made up of the input array and learning target, into two subsets, with 75% allocated for training and the remaining 25% for testing, following the methods of previous studies (e.g., Breiman, 2001; Guo et al., 2024). For a detailed account of the hyperparameter selections for each model, please refer to the Text S1 in supplementary materials. We find that RF provided the most robust predictions, as evidenced by higher correlations and smaller root mean square error (RMSE) values in most basins.

[Table 2]:
Table 2. Basic information on the comparison of the different model-derived with observed $V_{max}$ in Western Pacific (WP), North Atlantic (NA), North Indian (NI), South Indian (SI) South Pacific (SP) and Eastern Pacific (EP). CE, correlation

coefficients; RMSE, root mean square error. RF, random forecast; ANN, artificial neural network; CNN, convolutional neural network; SVR, support vector regressor; MLR, multivariate linear regression.

| | WP | NA | NI | SP | SI | EP |
|---|---|---|---|---|---|---|
| $RF_{CE}$ | 0.98 | 0.99 | 0.99 | 0.99 | 0.98 | 0.99 |
| $ANN_{CE}$ | 0.98 | 0.99 | 0.99 | 0.98 | 0.99 | 0.97 |
| $CNN_{CE}$ | 0.97 | 0.99 | 0.98 | 0.97 | 0.98 | 0.97 |
| $SVR_{CE}$ | 0.99 | 0.99 | 0.98 | 0.99 | 0.99 | 0.99 |
| $MLR_{CE}$ | 0.97 | 0.98 | 0.98 | 0.97 | 0.97 | 0.96 |
| $RF_{RMSE}$ (m/s) | 2.60 | 4.09 | 1.33 | 3.73 | 3.25 | 5.05 |
| $ANN_{RMSE}$ (m/s) | 5.09 | 5.31 | 1.65 | 3.87 | 4.37 | 10.05 |
| $CNN_{RMSE}$ (m/s) | 5.92 | 8.39 | 2.43 | 7.18 | 7.30 | 11.2 |
| $SVR_{RMSE}$ (m/s) | 3.99 | 6.70 | 2.18 | 4.87 | 5.03 | 9.08 |
| $MLR_{RMSE}$ (m/s) | 7.33 | 9.34 | 2.28 | 7.42 | 7.45 | 12.49 |

[lines 181-183]:

We also test several machine learning models (Table 3). Although the ANN-derived $R_{max}$ exhibit stronger correlations with observations, the RMSE values of $R_{max}$ derived by RF with observations are considerably smaller than that derived by other models.

[Table 3]:

Table 3. Similar to Table 2, but for $R_{max}$.

| | WP | NA | NI | SP | SI | EP |
|---|---|---|---|---|---|---|
| $RF_{CE}$ | 0.93 | 0.96 | 0.96 | 0.91 | 0.96 | 0.93 |
| $ANN_{CE}$ | 0.96 | 0.97 | 0.93 | 0.97 | 0.96 | 0.94 |
| $CNN_{CE}$ | 0.95 | 0.96 | 0.95 | 0.97 | 0.94 | 0.96 |
| $SVR_{CE}$ | 0.06 | 0.21 | 0.26 | 0.25 | 0.01 | 0.07 |
| $MLR_{CE}$ | 0.90 | 0.93 | 0.98 | 0.98 | 0.96 | 0.84 |
| $RF_{RMSE}$ (km) | 20.80 | 31.47 | 10.48 | 15.11 | 16.51 | 24.75 |
| $ANN_{RMSE}$ (km) | 31.96 | 46.74 | 16.62 | 21.06 | 23.22 | 41.14 |
| $CNN_{RMSE}$ (km) | 34.93 | 52.89 | 22.04 | 20.97 | 25.69 | 44.07 |
| $SVR_{RMSE}$ (km) | 43.53 | 72.43 | 28.26 | 29.05 | 30.99 | 51.15 |
| $MLR_{RMSE}$ (km) | 37.65 | 57.82 | 21.93 | 23.35 | 27.22 | 44.16 |

*2. One suggestion for improving writing could be to streamline the description of the wind profile models, as the detailed mathematical formulations might be overwhelming for some readers. Instead, focusing on the selected wind profile models and the comparative performance of all models in the main body of the manuscript (rather than in supplement and summarize the tables) would be more impactful.*

**Response:** Per your comment, we have streamlined the description of the wind profile models, and moved the detailed mathematical formulations to the supplementary materials. Besides, we have added the comparative performance of all models. For the revised statements please see lines 208-210 and lines 276-285.

[lines 208-210]:

We evaluate the performance of each profile model by comparing $R_{34}$, $R_{50}$, and $R_{64}$ estimates with those recorded in the IBTrACS dataset. Subsequently, we select the optimal model to generate reconstructed $R_{34}$, $R_{50}$, and $R_{64}$, as described in detail in Section 4.

[lines 276-285]:

In the WP basin, the W06 model demonstrates the strongest correlation ($R_{34}$: 0.89, $R_{50}$: 0.82, $R_{64}$: 0.78), achieving the lowest RMSE and MAE. In NA basin, the CLE15 model outperforms others for $R_{34}$, with a correlation coefficient of 0.87, RMSE of 78.77 km, and MAE of 53 km, whereas the W06 model performs better for $R_{50}$ and $R_{64}$. For the NI and SI basins, all models except W06 show poor correlation with observations, some even exhibiting negative correlations. In the SP and EP basins, W06 substantially surpasses other models in terms of correlation coefficient. Although other models produce slightly smaller RMSE and MAE values for $R_{64}$ in the EP basin compared to W06, their correlation coefficients, which are $< 0.2$, justify our choice of W06. Consequently, we select W06 to forecast $R_{34\_RC}$, $R_{50\_RC}$, and $R_{64\_RC}$ for the WP, NI, SI, SP and EP basins, whereas for the NA basin, we use CLE15 to predict $R_{34\_RC}$ and W06 to predict $R_{50\_RC}$ and $R_{64\_RC}$. The correlation coefficients are $>0.75$ for three outer size metrics in most basins (Table 6).

*3. The reductions in bias for key metrics like Vmax and Rmax are impressive. However, while the manuscript acknowledges the limitations related to landfall TCs and the dependency on ERA5's spatial resolution, a more detailed discussion on how these limitations might affect specific use cases of the dataset could be beneficial.*

**Response:** Thank you for this good comment. We have supplemented the detailed discussion. For the revised statements please see lines 295-297 and lines 35-59.

[lines 295-297]:

When employing this dataset for the purpose of examining the characteristics and impacts of TCs during their landfall, it is possible to overestimate their intensity while underestimating the scope of their influence.

[lines 301-302]:

Therefore, when assessing the impacts of TCs using this dataset, e.g., TC risk assessment, it is crucial to validate the results through observations from meteorological stations, buoys, and other relevant methods.

*4. Ensure consistency in tense usage, particularly when discussing results and implications. For example, there is a mixture of the past simple tense and present simple tense in line 240.*

**Response:** Per your comment, we have consistently applied the present simple tense throughout the revised manuscript. We have revised the sentences in line 240 as "Therefore, for clarity, the $R_{max\ \_ERA5}$ data are not shown with the reconstructed TC

results in Fig. 5. The MAE exhibits a reduction of 39.57 km on a global scale, with a further reduction of over 59.37 km in the SI basin, as described in detail in Table 5."

*5. Consider using active voice more frequently to make the writing more direct. For example, "Six wind profile models were used to compute the radii..." could be "We used six wind profile models to compute the radii..."*

**Response:** Per your kind comments, we have revised the verb tenses in the revised manuscript as active voice, and have revised the sentence as "We use six wind profile models to compute the radii to locations with sustained wind speeds of 34, 50, and 64 knots ($R_{34}$, $R_{50}$, and $R_{64}$), and the selected wind profile models (CLE15 for $R_{34}$ in the North Atlantic, W06 for others) show good estimates for TC outer sizes, with correlation coefficients $> 0.75$ for three outer size metrics in most basins."

---

## Author Comment (AC2)

**Responses to comments of Reviewer #2**

We wish to express our great appreciation for your thoughtful review and constructive comments. In the revised manuscript, we have made substantial changes to strengthen its readability, and incorporated all your comments. In the response below, we address each of these comments individually. Your comments are *italicized* and our responses follow immediately. Changes to the manuscript text are included in blue.

*In Xu et al. (2024), the authors introduce an extension of the International Best Track Archive for Climate Stewardship (IBTrACS) dataset. Using the ERA-5 reanalysis dataset alongside IBTrACS observations, the authors utilize random forest to generate estimates of maximum wind speed and radius of maximum wind. The authors then proceed to construct minimum pressure estimates using an empirical wind-pressure relationship and the newly generated maximum wind speed estimates, before incorporating the radius of maximum wind and maximum wind speed estimates into wind models to generate estimates of the surface wind radii (radius of 17, 26, and 33 ms-1 winds). Validation statistics suggest that these estimates are more accurate than ERA-5 reanalysis data by itself, and the overall number of estimates generated in this manuscript provides substantially more data to be used by climatologists.*

*Overall, this paper presents an interesting and novel approach to handling missing observations in the tropical cyclone record. However, I have some concerns regarding how the dataset was created and how the manuscript is constructed. As such, I believe major revisions are necessary before this manuscript can be accepted.*

**Major Comments:**

*1. The selection of random forest as the primary method for this paper is not presented in a way that convinces the readers that it was the best choice. In Lines 146–148, the authors list several machine learning techniques that could be used, but random forest is chosen without demonstrating how it outperformed the other methods. In Lines 161–162, the hyperparameters of each random forest are noted to have been selected by "randomized search." This implies that the authors picked random numbers until the output was deemed acceptable, instead of utilizing validation statistics. Additionally, the authors do not mention the software or program used to implement the random forest (i.e., was it done using R, Python, SAS, Matlab, etc.), nor do they inform the reader of the error rate when training the random forest. By incorporating more details into how random forest outperformed the other potential techniques, how the best hyperparameters were determined, and how the process was conducted would go a long way to reassuring readers that the data they are using were generated using best practices.*

**Response:** Thank you for this good comment. We would address this comment in point-by-point response as follows.

*In Lines 146–148, the authors list several machine learning techniques that could be used, but random forest is chosen without demonstrating how it outperformed the other methods.*

We have added the evaluation of other methods in the section 3.2, and rewritten most parts of this section. For the revised statements please see lines 162-170, Table 2, lines 182-183 and Table 3.

[lines 162-170]:

We optimize the machine learning models by Randomized Search Cross-Validation with mean square error as the loss function using Python. The models include a random forest (RF) algorithm, artificial neural network (ANN), convolutional neural network, support vector regressor, and multivariate linear regression (Table 2). In the above-mentioned models, we incorporate data for the entire period (1959–2022) into the model training process. We randomly divide the dataset, made up of the input array and learning target, into two subsets, with 75% allocated for training and the remaining 25% for testing, following the methods of previous studies (e.g., Breiman, 2001; Guo et al., 2024). For a detailed account of the hyperparameter selections for each model, please refer to the Text S1 in supplementary materials. We find that RF provided the most robust predictions, as evidenced by higher correlations and smaller root mean square error (RMSE) values in most basins.

[Table 2]:
Table 2. Basic information on the comparison of the different model-derived with observed $V_{max}$ in Western Pacific (WP), North Atlantic (NA), North Indian (NI), South Indian (SI) South Pacific (SP) and Eastern Pacific (EP). CE, correlation coefficients; RMSE, root mean square error. RF, random forecast; ANN, artificial neural network; CNN, convolutional neural network; SVR, support vector regressor; MLR, multivariate linear regression.

| | WP | NA | NI | SP | SI | EP |
|---|---|---|---|---|---|---|
| RF$_{CE}$ | 0.98 | 0.99 | 0.99 | 0.99 | 0.98 | 0.99 |
| ANN$_{CE}$ | 0.98 | 0.99 | 0.99 | 0.98 | 0.99 | 0.97 |
| CNN$_{CE}$ | 0.97 | 0.99 | 0.98 | 0.97 | 0.98 | 0.97 |
| SVR$_{CE}$ | 0.99 | 0.99 | 0.98 | 0.99 | 0.99 | 0.99 |
| MLR$_{CE}$ | 0.97 | 0.98 | 0.98 | 0.97 | 0.97 | 0.96 |
| RF$_{RMSE}$ (m/s) | 2.60 | 4.09 | 1.33 | 3.73 | 3.25 | 5.05 |
| ANN$_{RMSE}$ (m/s) | 5.09 | 5.31 | 1.65 | 3.87 | 4.37 | 10.05 |
| CNN$_{RMSE}$ (m/s) | 5.92 | 8.39 | 2.43 | 7.18 | 7.30 | 11.2 |
| SVR$_{RMSE}$ (m/s) | 3.99 | 6.70 | 2.18 | 4.87 | 5.03 | 9.08 |
| MLR$_{RMSE}$ (m/s) | 7.33 | 9.34 | 2.28 | 7.42 | 7.45 | 12.49 |

[lines 182-183]:

We also test several machine learning models (Table 3). Although the ANN-derived $R_{max}$ exhibit stronger correlations with observations, the RMSE values of $R_{max}$ derived by RF with observations are considerably smaller than that derived by other models.

Table 3. Similar to Table 2, but for $R_{max}$.

|  | WP | NA | NI | SP | SI | EP |
|---|---|---|---|---|---|---|
| $RF_{CE}$ | 0.93 | 0.96 | 0.96 | 0.91 | 0.96 | 0.93 |
| $ANN_{CE}$ | 0.96 | 0.97 | 0.93 | 0.97 | 0.96 | 0.94 |
| $CNN_{CE}$ | 0.95 | 0.96 | 0.95 | 0.97 | 0.94 | 0.96 |
| $SVR_{CE}$ | 0.06 | 0.21 | 0.26 | 0.25 | 0.01 | 0.07 |
| $MLR_{CE}$ | 0.90 | 0.93 | 0.98 | 0.98 | 0.96 | 0.84 |
| $RF_{RMSE}$ (km) | 20.80 | 31.47 | 10.48 | 15.11 | 16.51 | 24.75 |
| $ANN_{RMSE}$ (km) | 31.96 | 46.74 | 16.62 | 21.06 | 23.22 | 41.14 |
| $CNN_{RMSE}$ (km) | 34.93 | 52.89 | 22.04 | 20.97 | 25.69 | 44.07 |
| $SVR_{RMSE}$ (km) | 43.53 | 72.43 | 28.26 | 29.05 | 30.99 | 51.15 |
| $MLR_{RMSE}$ (km) | 37.65 | 57.82 | 21.93 | 23.35 | 27.22 | 44.16 |

*In Lines 161–162, the hyperparameters of each random forest are noted to have been selected by "randomized search." This implies that the authors picked random numbers until the output was deemed acceptable, instead of utilizing validation statistics. Additionally, the authors do not mention the software or program used to implement the random forest (i.e., was it done using R, Python, SAS, Matlab, etc.), nor do they inform the reader of the error rate when training the random forest.*

We have also supplemented a detailed account of the hyperparameters selections for each model in supplementary materials. The software used to implement the mechine learning models is Python, and the related description has been added as " We optimize the machine learning models by Randomized Search Cross-Validation with mean square error as the loss function using Python". The error rates when training the random forest are also added in the section 3.2. Because we opt for the 'neg_mean_ squared_error' as the loss function, the program can not give error rates directly. Therefore, we define the error rate of the random forest on the training data as the absolute relative errors between the predicted and observed data, normalized by the observations. For the revised statements please see Text S1 in supplementary materials, lines 173-176 and lines 187-188.

**Text S1. The hyperparameter selections for each model**
**Text S1.1 Random forecast**

When optimizing the random forecast model for regression tasks, we utilize Randomized Search Cross-Validation (RandomizedSearchCV) to systematically explore a wide range of hyperparameters.

Specifically, we define hyperparameter distributions that encompass a range of values for the number of trees in the forest (100 to 300), maximum depth of the tree (10 to 30), maximum number of features (1 to 7), minimum number of samples (2 to 10), minimum number of samples (2 to 20), and maximum number of leaf nodes (800 to 1200).

With 500 iterations and 5-fold cross-validation, we search for the optimal hyperparameter combination that minimized the mean squared error (MSE), which is a common choice for regression problems due to its ability to penalize large errors. Leveraging parallel computing, we efficiently fit the model to the training data and obtain the best-performing estimator.

**Text S1.2 Support vector machine**

Similarly, we utilize RandomizedSearchCV with an extensive hyperparameter grid to optimize a support vector machine (SVM) regression model.

This grid comprises key parameters such as the regularization parameter C, ranging from 1.0 to 1000, to strike a balance between training error and margin. We explore the kernel, encompassing radial basis function, polynomial, and sigmoid options, with each option influencing the model's decision function. Further, we adjust the influence of each training sample on the decision function through the gamma parameter, which varies from 0.001 to 1. The epsilon parameter (0.01 to 0.1) defines the epsilon-insensitive zone, thereby controlling the margin width.

Leveraging 500 iterations and 5-fold cross-validation, we systematically search for the optimal hyperparameter combination that minimized the MSE, thereby refining the SVM regression model for enhanced performance.

**Text S1.3 Artificial neural network**

We also optimize the artificial neural network model by RandomizedSearchCV in our research to tackle the regression task, featuring a series of densely interconnected layers.

For the number of hidden layers, we explore between 1 and 4 layers to find the optimal layer depth that could capture the complexity of the data without causing overfitting. We adjust the size of each layer (number of neurons) between 1 and 500 to optimize the model's expressive power and generalization ability. The Adam optimization algorithm initializes with a learning rate through random search using reciprocal distribution within a wide range (from 0.0001 to 0.01). Across all layers, we employ the rectified linear unit activation function to introduce non-linearity.

To optimize the model's performance, we opt for the MSE as the loss function. During the training phase, the model undergoes rigorous training leveraging 500 iterations and 5-fold cross-validation, ensuring optimal performance and robustness.

**Text S1.4 Convolutional neural network**

For performance evaluation of the convolutional neural network, we adopt the MSE loss function. We also optimize the model by RandomizedSearchCV.

Leveraging 500 iterations and 5-fold cross-validation, the model undergoes rigorous training on the designated training dataset. Specifically, we define a range of values for each hyperparameter (e.g., dropout rate between 0.001 and 0.5, learning rate between 0.0001 and 0.01, filter size between 2 and 5, number of filters between 32 and 128, and number of dense units between 64 and 256). The Adam optimizer dynamically adjusts the learning rate throughout training, further enhancing the optimization process.

**Text S1.5 Multivariate linear regression**

We also apply a multivariate linear regression model to investigate the relationship between the predictor variables and the response variable. We train the multiple linear

regression model on the training set to estimate the coefficients of the linear relationship, which is subsequently used to predict the response variable on the test set.

[lines 173-176]:

To further assess the accuracy of the RF model, we define the error rate of the RF on the training data as the absolute relative errors between the predicted and observed $V_{max}$, normalized by the observations. The error rates are 0.11, 0.16, 0.09, 0.19, 0.16 and 0.20 for the WP, North Atlantic (NA), North Indian (NI), South Indian (SI), South Pacific (SP) Eastern Pacific (EP) and basins, respectively.

[lines 187-188]:

In the RF models, the error rates are 0.19, 0.23, 0.14, 0.19, 0.15 and 0.23 for the WP, NA, NI, SI, SP and EP basins, respectively.

**Minor Comments and Technical Corrections:**

*1. Lines 10–11: The first two sentences of the abstract have different verb tenses, with the first sentence written in active voice while the second sentence is written in passive voice. This occurs throughout the paper. Please be consistent, preferably in active voice.*

**Response:** Per your kind comments, the verb tenses in the manuscript and supplementary materials have been revised as active voice. The first two sentences of abstract have been revised as " Tropical cyclones (TCs) are powerful weather systems that can cause extreme disasters. The International Best Track Archive for Climate Stewardship (IBTrACS) dataset provides the widely used data to estimate TC climatology. "

*2. Line 18: The abstract mentions the authors used "10 m azimuthal median azimuthal wind profiles," however in Line 118 the authors mention using "10 m azimuthal mean azimuthal wind profiles." These are not the only instances where the wind profiles differ. Please adjust to state the correct wind profile (mean or median).*

**Response:** We have corrected "10 m azimuthal median azimuthal wind profiles" as "10 m azimuthal mean azimuthal wind profiles ".

*3. Line 26: Referring to something as "significant" typically implies statistical significance. Unless you have statistics to support the claim, consider using a similar word, such as "substantial."*

**Response:** Per your comment, we revised it as " substantial".

*4. Lines 26–27: Words like "formidable," "torrential," and "devastating" are not typically used in scientific writing. Please adjust accordingly.*

**Response:** We revised the sentence as " Tropical cyclones (TCs) are powerful weather systems accompanied by gale winds, heavy rainstorms, substantial waves, and severe storm surges, which cause extensive damage in affected regions (Gray, 1968)."

*5. Line 28: Are the 29 billion dollars in damages presented here in US dollars? If so, which year?*

**Response:** The 29 billion dollars in damages presented here are in US dollars, and the time period is from 1996-2016. To demonstrate the most recent statistics, we consulted the Emergency Events Database report for the year 2023, and revised the sentence as " During the 2003-2022 period, the global average of TCs is 104 annually, resulting in estimated annual economic losses of 95.6 billion US dollars and affecting more than 3.2 million individuals".

*6. Line 28: Are the 22 million individuals globally distributed, or located on a specific continent?*

**Response:** The 22 million individuals are globally distributed. We have added it in the revised sentence in the comment 5.

*7. Line 29: The previous sentence establishes the scale of TC-related disasters, but the frequency has not been addressed. Please consider adding information pertaining to TC frequency.*

**Response:** During the 2003-2022 period, the global average of TCs is 104 annually. We have added it in the revised sentence in the comment 5.

*8. Line 50: Consider changing "radius to maximum wind speed" to "radius of maximum wind," as this aligns with other manuscripts. This appears elsewhere in the manuscript, not just here.*

**Response:** Per your kind comment, we have revised it.

*9. Line 52: Consider changing "on surface" to "near the surface."*

**Response:** Per your comment, we have revised it.

*10. Lines 71–73: It would be helpful to add a couple of sentences to demonstrate to readers how reanalysis can be used to reconstruct TC size. Some examples:*
*Gori et al., 2023: North Atlantic Tropical Cyclone Size and Storm Surge Reconstructions From 1950-Present.*
*Schenkel et al., 2017: Evaluating Outer Tropical Cyclone Size in Reanalysis Datasets Using QuikSCAT Data.*
*Thompson et al., 2024: Construction of a tropical cyclone size dataset using reanalysis data.*
*Zick and Matyas, 2016: Tropical cyclones in the North American regional reanalysis: the impact of satellite-derived precipitation over ocean.*
**Response:** Per your comment, we have added the sentences on how reanalysis can be

used to reconstruct TC size. For the revised statements please see lines 71-76.

[lines 71-76]:

Schenkel et al. (2017) evaluates whether reanalysis dataset can be used to derive a long-term TC size dataset utilizing QuikSCAT data. Zick and Matyas (2016) explore the impact of satellite-derived precipitation over ocean on TC in the North American Regional Reanalysis. Gori et al. (2023) uses ERA5 reanalysis data to estimate the TC outer size, and wind model to estimate the radius of maximum wind. Thompson et al. (2024) constructs a tropical cyclone (TC) size dataset using the NCEP/NCAR Reanalysis I dataset for landfalling TCs along the United States coastline from 1948 to 2022.

*11. Line 78: Consider re-wording "relatively accurate TC intensity," as this implies that ERA-5 intensity is accurate (the description of maximum wind biases later in the manuscript undermines this phrasing).*

**Response:** Per your comment, we have revised it as " studies that have employed it to derive relatively accurate TC intensity data are lacking ".

*12. Line 79: In the abstract, it says that this paper is about a reconstructed dataset. Here, it says a new dataset is generated. Please adjust throughout the manuscript for consistency.*

**Response:** Per your comment, we have revised "a new TC dataset" as " a reconstructed TC dataset" in the manuscript.

*13. Lines 91–92: IBTrACS can be downloaded in netCDF, CSV, and Shapefile format. Which version did you use?*

**Response:** We used the version of netCDF format. We have added the format as " Data on TC tracks, intensity, and size are obtained from the IBTrACS (version 4r01 in netCDF format) ".

*14. Line 93: Which U.S. agencies were responsible for the data?*

**Response:** The National Oceanic and Atmospheric Administration's (NOAA) National Hurricane Center (NHC) was responsible for the data concerning the North Atlantic and East Pacific. The military's Joint Typhoon Warning Center (JTWC) for the were responsible for the data concerning the remainder of the globe. We have added related information. For the revised statements please see lines 98-100.

[lines 98-100]:

As the TC $R_{max}$ data from all main TC basins are accessible from U.S. agencies (the National Oceanic and Atmospheric Administration's National Hurricane Center for the North Atlantic and East Pacific and the military's Joint Typhoon Warning Center

for the remainder of the globe), we employ these data and exclude the irregular time steps.

*15. Line 118: You cite Schenkel (2017), but your references do not contain a paper where Schenkel is the solo author. Was this supposed to be Schenkel et al. (2017), or did you intend Schenkel (2017) but forget to add the citation (Schenkel did publish a solo author paper in 2017 pertaining to TC centers, which is the topic of the sentence)? Please revisit your citations to be sure that all in-text citations and references align.*

**Response:** This was supposed to be Schenkel et al. (2017), and we have corrected it.

*16. Lines 120–122: How was this done? Which software did you use?*

**Response:** We obtain the centers of six reanalysis variables (SLP; relative vorticity at 700, 850, and 925 hPa; and geopotential height at 700 and 850 hPa) by calculating the centroids of positive relative vorticity values and negative other variables values over the grid near the first guess position (±2° ) using Python. Subsequently, the centers are averaged to adjust the position of the estimated reanalysis TC center. We have added related it. For the revised statements please see lines 126-130.

[lines126-130]:
    To remove uncertainties associated with TC centers in the reanalysis data, we obtain the centers of six reanalysis variables (SLP; relative vorticity at 700, 850, and 925 hPa; and geopotential height at 700 and 850 hPa) by calculating the centroids of positive relative vorticity values and negative other variables values over the grid near the first guess position (±2° ) using Python. Subsequently, we average the centers to adjust the position of the estimated reanalysis TC center.

*17. Lines 136–137: What is causing this systemic bias in ERA-5 derived maximum wind? An earlier sentence mentioned a couple of reasons, but it is presented as being an issue for other datasets. Is it the same reasoning, or are there different reasons for this bias?*

**Response:** We consulted the ECMWF Technical Memoranda "Tropical cyclone activities at ECMWF", and found that the lower $P_{min}$ and the underestimation of the TC wind-pressure relation described in ERA5 may cause this systemic bias. Besides, $V_{max}$ is substantially influenced by convective-scale processes that are not adequately represented in global models, leading to an inherent tendency for underestimation. We have added it. For the revised statements please see lines 143-146.

[lines143-146]:
    The biases of $V_{max}$ are less dependent on the basin, suggesting the systematic underestimation of $V_{max}$ by the ERA5 data, partly due to the lower $P_{min}$ and the underestimation of the TC wind-pressure relation described in ERA5 (Magnusson et al., 2021). Moreover, convective-scale processes substantially influence $V_{max}$, which

cannot be adequately represented in global models, leading to an inherent tendency for underestimation.

*18. Line 138: Different basins use different scales to categorize TCs. Why did you select the Saffir-Simpson Hurricane Wind Scale for all basins?*

**Response:** Despite the fact that different basins use different scales to categorize TCs. It is our contention that the adoption of a uniform scale would serve to more clearly demonstrate the performance of both ERA5 and model-derived data in different basins. In many previous studies on global TCs, researchers have adopted the Saffir-Simpson Hurricane Wind Scale. Some examples:

Wright C J, 2019: Quantifying the global impact of tropical cyclone-associated gravity waves using HIRDLS, MLS, SABER and IBTrACS data.

Bloemendaal N et al., 2022: A globally consistent local-scale assessment of future tropical cyclone risk.

Mo Y et al., 2023: Tropical cyclone risk to global mangrove ecosystems: potential future regional shifts.

Pérez-Alarcón A et al., 2021: Comparative climatology of outer tropical cyclone size using radial wind profiles.

Therefore, we select the Saffir-Simpson Hurricane Wind Scale for all basins. To clarify the reason, we have added the explanation. For the revised statements please see lines 146-149.

[lines 146-149]:

    To further demonstrate the performance of ERA5-derived data, we select the Saffir-Simpson categories as the uniform scale for all the basins, and analyze the differences between ERA5-derived and observed data across various wind speed ranges, following the methods in previous researches (Wright, 2019; Bloemendaal et al., 2022; Mo et al., 2023).

*19. Line 142: I am not sure "modesty" is the appropriate word here. A possible change to the sentence: "Despite the discrepancy in TC intensity, Bian et al. (2021) demonstrated that ERA-5 accurately depicts TC structural alterations."*

**Response:** Thanks for your suggestion**.** We have revised the sentence as " Despite the discrepancy in TC intensity, Bian et al. (2021) demonstrates that ERA-5 accurately depicts TC structural alterations."

*20. Line 166: Which correlation did you use (Pearson, Spearman, or Kendall)? How was statistical significance evaluated?*

**Response:** We use Pearson correlation coefficients. The statistical significance of Pearson correlation coefficients is evaluated through the application of a t-test. We have added it in the manuscript. For the revised statements please see lines 188-191.

[lines 188-191]:
    We further evaluate model performance by comparing the model-derived and observed $V_{max}$ and $R_{max}$ on the testing dataset in Section 4, using a comprehensive set of statistical metrics, including mean error, mean absolute error (MAE), RMSE, and Pearson correlation coefficients. We evaluate the statistical significance of Pearson correlation coefficients through the application of a t-test.

*21. Lines 180–201: Are the same variables incorporated into each wind profile? For example, is the distance r in Holland (1980) the same r in DeMaria (1987)?*

**Response:** Yes, the same variables are incorporated into each wind profile. To make it clear, we have added the related description. For the revised statements please see lines 59-61 in the supplementary meterials.

[lines 59-61 in supplementary meterials]:
    In the models, $r$, $R_{max}$ and $V_{max}$ represent the distance from the cyclone center, radius of maximum wind speed and maximum sustained wind speed, respectively.

*22. Line 187: What are R1 and R2?*

**Response:** $R_1$ and $R_2$ are the location of the eye and the location of the transition zone, respectively. To make it clear, we have added the detailed description. For the revised statements please see lines 68-76 in the supplementary meterials.

[lines 68-76 in supplementary meterials]:
    The model proposed by Willoughby et al. (2006; hereinafter, W06) is formulated as follows:

$$V(r) = \begin{cases} V_i = V_{max}\left(\frac{r}{R_{max}}\right)^n, & 0 \le r \le R_1 & \text{(S3a)} \\ V_i(1-w) + V_0 w, & R_1 \le r \le R_2 & \text{(S3b)} \\ V_0 = V_{max}e^{-\frac{r-R_{max}}{X_1}}, & R_2 \le r & \text{(S3c)} \end{cases}$$

where $V_i$ and $V_0$ are the tangential wind components in the eye and beyond the transition zone, which lies between $r = R_1$ and $r = R_2$, respectively. The transition zone is defined as the radius of maximum wind from the cyclone inner to outer profiles. $w$, $X_1$, and $n$ are the weight function, exponential decay length in the outer vortex, and power law exponent within the eye, respectively. $R_2$ is the location of the transition zone, and determined by requiring the radial derivative of (S3c) to vanish at $r = R_{max}$. $R_1$ is the location of the eye, and can be solved as follows:

$$w(\xi) = 126\xi^5 - 420\xi^6 + 540\xi^7 - 315\xi^8 + 70\xi^9 \tag{S3d}$$

$$w\left(\frac{R_{max}-R_1}{R_{max}}\right) = \frac{nX_1}{nX_1+R_{max}} \tag{S3e}$$

*23. Line 188: What is the transition zone?*

**Response:** The transition zone is defined as the radius of maximum wind from the cyclone inner to outer profiles. The determination of its location are presented in the comment 22.

*24. Line 216: Providing the error rate of the random forest on the training data would also help with assessing the accuracy of the reconstructed data.*

**Response:** Because we opt for the "neg_mean_ squared_error" as the loss function, the program can not give error rates directly. Therefore, we define the error rate of the random forest on the training data as the absolute relative errors between the predicted and observed data, normalized by the observations. For the revised statements please see lines 173-176 and lines 187-188.

[lines 173-176]:
    To further assess the accuracy of the RF model, we define the error rate of the RF on the training data as the absolute relative errors between the predicted and observed V_{max}, normalized by the observations. The error rates are 0.11, 0.16, 0.09, 0.19, 0.16 and 0.20 for the WP, North Atlantic (NA), North Indian (NI), South Indian (SI), South Pacific (SP) Eastern Pacific (EP) and basins, respectively.

[lines 187-188]:
    In the RF models, the error rates are 0.19, 0.23, 0.14, 0.19, 0.15 and 0.23 for the WP, NA, NI, SI, SP and EP basins, respectively.

*25. Lines 217–218: According to the flowchart in Figure 3, the observational data from IBTrACS were used to generate the reconstructed versions of the same data. This would explain why you have such high correlations. In practice, you typically do not see such high correlation coefficients, and that is cause for concern. In the abstract, it says that the observation data were used for training. Were they also used in the test data?*

**Response:** We apologize for not making it clear how we evaluated the model's performance. The observational data were also used in the test data, but it should be noted that the data within the training and test datasets are not identical. Besides, the evaluation of model's performance in Section 4, including the correlations, was conducted on the testing data. The ERA5-derived and observed data served as the input array and learning target, respectively. We randomly divided the dataset, made up of the input array and learning target, into two subsets, with 75% allocated for training and the remaining 25% for testing.

    The high correlations between model-derived and observed data are likely attributable to the strong correlation coefficients between the ERA5-derived and observed data, with the correlation coefficients of $V_{max}$ larger than 0.9 for all the basins and $R_{max}$ larger than 0.7 in Western Pacific, North Indian, South Indian, South Pacific basins. In North Atlantic and Eastern Pacific, the correlations between $R_{max}$

in IBTrACS and in ERA5 are low, which leads to larger error bars in the modelled results.

We have revised the sentence in the abstract as "…, with $V_{max}$ and $R_{max}$ data from the IBTrACS dataset used as learning target data. "

We have also added the sentence in the description of flowchart in Figure 3 as "We evaluate model performance by comparing the model-derived and observed $V_{max}$ and $R_{max}$ on the testing dataset, using a comprehensive set of statistical metrics. "

*26. Line 225: While the reconstructed data do appear to better fit the observational data, the error bars are quite large and the reconstructed data are estimates and therefore not "real" data. Saying they are "clearly superior" is a stretch.*

**Response:** Thanks for your suggestion. We have revised the sentence as "…, and thus produced better predictions for all six basins. "

*27. Line 233: Does Table 2 compare the ERA-5 and reconstructed data with the observation data? If so, please make that clear in the table caption.*

**Response:** Table 2 does compare the ERA-5 and reconstructed data with the observation data. We have revised it as " Basic information on the comparison of the ERA5-derived and reconstructed with observed $V_{max}$"

*28. Line 246: Why are the error bars so much larger for the North Atlantic and East Pacific basins compared to the other basins?*

**Response:** The larger error bars for North Atlantic and East Pacific may be attributed to the low correlations between $R_{max}$ in IBTrACS and in ERA5 (North Atlantic: 0.37; North Atlantic: -0.02). We have added the explanation as " It is noteworthy that the error bars are larger for the NA and EP basins in comparison to the other basins. This may be attributed to the low correlations between $R_{max}$ in IBTrACS and in ERA5 (NA: 0.37; EP: -0.02). "

*29. Line 247: For each basin, the MAE of the reconstructed Rmax is less than the MAE for ERA-5, but at the global level, MAE for the reconstructed Rmax is greater than ERA-5. It looks like the two numbers were switched. Is that the case?*

**Response:** Thanks for your correction. We have corrected it.

*30. Line 258: What is the "sea level pressure difference"? It isn't defined anywhere else in the paper.*

**Response:** "sea level pressure difference" is the difference between environmental pressure and typical cyclone minimum central pressure. We have revised it.

*31. Line 268: Consider adding ", respectively." after the second sentence.*

**Response:** Per you comment, we have added it.

*32. Line 287: Consider "Finally" instead of "Besides"*

**Response:** Per you comment, we have revised it.

*33. Lines 294–295: In the Zenodo repository, consider adding a ReadMe file that explains what the columns are in the dataset (how they were derived, their units, etc.). This would provide helpful information for users who would use your data.*

**Response:** Per you comment, we have added a ReadMe file in the Zenodo repository.

---

## Author Response (AR2)

**Responses to comments of Reviewer #2**

We wish to express our great appreciation for your review and valuable technical corrections. In the revised manuscript, we have made the correction, and incorporated all your comments. In the response below, we address each of these comments individually. Your comments are *italicized* and our responses follow immediately. Changes to the manuscript text are included in blue.

*Thank you for incorporating the original recommendations. At this point, only a few minor technical corrections need to be addressed, which are listed below.*

*1.  Line 11: …"dataset provides the widely used data" – delete "the" here.*

**Response:** Thank you for this correction. We have deleted "the" in this sentence.

*2.  Line 62: Change "researches" to "studies"*

**Response:** We have revised " researches" as " studies".

*3.  Line 76: Change "dataset" to "datasets"*

**Response:** We have revised it.

*4.  Line 78: Change "TC" to "TCs"*

**Response:** We have revised it.

*5. Line 79: "…ERA5 reanalysis data to estimate the TC outer size…" = either delete "the" or make TC "TCs"*

**Response:** We have revised TC as "TCs".

*6. Line 112: "…IBTrACS dataset only records…" – delete "dataset" here.*

**Response:** We have deleted " dataset" in this sentence.

*7. Line 166: Change "researches" to "research"*

**Response:** We have revised it.

*8. Lines 203–204: Add the abbreviations for convolutional neural network, support vector regressor, and multivariate linear regression.*

**Response:** We have added the abbreviations. The sentence has been revised as " The models include a random forest (RF) algorithm, artificial neural network (ANN), convolutional neural network (CNN), support vector regressor (SVR), and multivariate linear regression (MLR), as detailed in Table 2."

*9. Line 387: "In NA basin, the CLE15 model…" – add "the" between In and NA.*

**Response:** We have added it.